# TWIST1 Upregulation Is a Potential Target for Reversing Resistance to the CDK4/6 Inhibitor in Metastatic Luminal Breast Cancer Cells

**DOI:** 10.3390/ijms242216294

**Published:** 2023-11-14

**Authors:** Nicoletta Cordani, Luca Mologni, Rocco Piazza, Pietro Tettamanti, Viola Cogliati, Mario Mauri, Matteo Villa, Federica Malighetti, Camillo Di Bella, Marta Jaconi, Maria Grazia Cerrito, Guido Cavaletti, Marialuisa Lavitrano, Marina Elena Cazzaniga

**Affiliations:** 1School of Medicine and Surgery, Milano-Bicocca University, 20900 Monza, Italy; luca.mologni@unimib.it (L.M.); rocco.piazza@unimib.it (R.P.); pietro.tettamanti@unimib.it (P.T.); mario.mauri@unimib.it (M.M.); matteo.villa@unimib.it (M.V.); federica.malighetti@unimib.it (F.M.); mariagrazia.cerrito@unimib.it (M.G.C.); guido.cavaletti@unimib.it (G.C.); marialuisa.lavitrano@unimib.it (M.L.); marina.cazzaniga@unimib.it (M.E.C.); 2Phase 1 Research Centre, Fondazione IRCCS San Gerardo dei Tintori, 20900 Monza, Italycamillo.dibella@irccs-sangerardo.it (C.D.B.); marta.jaconi@irccs-sangerardo.it (M.J.)

**Keywords:** CDK4/6 inhibitors, Metastatic Luminal Breast Cancer, TWIST1, EMT

## Abstract

Cyclin-dependent kinase (CDK) 4/6 inhibitors have significantly improved progression-free survival in hormone-receptor-positive (HR+), human-epidermal-growth-factor-receptor-type-2-negative (HER2−) metastatic luminal breast cancer (mLBC). Several studies have shown that in patients with endocrine-sensitive or endocrine-resistant LBC, the addition of CDK4/6 inhibitors to endocrine therapy significantly prolongs progression-free survival. However, the percentage of patients who are unresponsive or refractory to these therapies is as high as 40%, and no reliable and reproducible biomarkers have been validated to select a priori responders or refractory patients. The selection of mutant clones in the target oncoprotein is the main cause of resistance. Other mechanisms such as oncogene amplification/overexpression or mutations in other pathways have been described in several models. In this study, we focused on palbociclib, a selective CDK4/6 inhibitor. We generated a human MCF-7 luminal breast cancer cell line that was able to survive and proliferate at different concentrations of palbociclib and also showed cross-resistance to abemaciclib. The resistant cell line was characterized via RNA sequencing and was found to strongly activate the epithelial-to-mesenchymal transition. Among the top deregulated genes, we found a dramatic downregulation of the CDK4 inhibitor CDKN2B and an upregulation of the TWIST1 transcription factor. TWIST1 was further validated as a target for the reversal of palbociclib resistance. This study provides new relevant information about the mechanisms of resistance to CDK4/6 inhibitors and suggests potential new markers for patients’ follow-up care during treatment.

## 1. Introduction

Breast cancer (BC) is the most frequent tumor in women and represents a leading cause of cancer-specific mortality worldwide [1]. In 2020, the World Health Organization’s International Agency for Research on Cancer (IARC) estimated 2.26 million new cases of BC [2]. Hormone-receptor-positive (HR+)/human-epidermal-growth-factor-receptor-type-2-negative (HER2−) BC is the most frequent subtype, representing about 70% of all BCs [3]. It has been estimated that approximately 20–30% of the patients affected by this tumor will experience relapse, developing metastatic disease. The median overall survival of HR+/HER2− metastatic BC is 46 months [4].

The cell cycle is regulated by cyclins and their partners, cyclin-dependent kinases (CDKs), and in this context, the CDK4/6–Cyclin D complex allows the cell cycle to progress from the G1 to S phases through the phosphorylation of the retinoblastoma protein. Preclinical data on cancer cell panels demonstrated that the inhibition of CDK4/6–Cyclin D complex promotes G1 arrest, which leads to senescence. Several clinical studies showed that the addition of CDK4/6 inhibitors to endocrine therapy results in a significant prolongation of progression-free survival (PFS) in metastatic LBC patients. This result was consistent across all subgroups of patients enrolled in the above-mentioned studies [5,6,7].

The use of CDK4/6 inhibitors (palbociclib, abemaciclib, or ribociclib) in association with endocrine therapy [8] has led to important changes in clinical practice; in fact, these treatments are currently proposed as first-line therapies for all metastatic BC patients without visceral crisis. However, the percentage of patients unresponsive to these therapies is as high as 43.7% [9], and despite the success of CDK4/6 inhibitors in the management of LBC, the occurrence of drug resistance is still a major problem to deal with. Unfortunately, no reliable and reproducible biomarker able to select a priori responders has been validated so far; therefore, determining the molecular features associated with resistance to these inhibitors is crucial. Recently, Yang C et al. established abemaciclib-resistant cell lines, identifying the amplification of CDK6 as a recurrent event [10].

The aim of the present in vitro study was the characterization of the genes associated with CDK4/6 inhibitor (CDK4/6i) resistance. We induced resistance to the CDK4/6i palbociclib by chronically exposing the HR+ cell line MCF-7 to increasing concentrations of the compound. This cell line was used as a model to identify genes associated with resistance to CDK4/6i and to understand how such alterations can modify cyclin D-CDK4/6-mediated growth control.

EMT genes, which emerged upregulated in our model resistant cell line, are known to induce a variety of complementary tumor characteristics including tumor cell stemness, tumorigenicity, resistance to therapy, and adaptation to microenvironmental changes [11]. TWIST1 is expressed in a large number of tumor types and has been shown to be a key regulator of breast cancer metastasis [12]. Indeed, by activating several target genes, including Bmi1, ZEB2, Snail1 and Snail2, which increase cell dedifferentiation and mobility, it is known to promote EMT and cell invasion [11]. As Twist1 was found to be over-expressed in palbociclib-resistant MCF-7 cells, we focused on it as an interesting therapeutic target to overcome CDK4/6i resistance.

## 2. Results

### 2.1. Generation of Palbociclib-Resistant MCF-7 Cells

To investigate the mechanisms of acquired resistance in patients affected by HR+ HER2− mLBC cells, we treated MCF-7 cells with gradually increasing concentrations of palbociclib to obtain resistant cells. After 12 months of continuous exposure to the drug, the MCF-7 cells acquired the ability to grow in the presence of 1 µM palbociclib. To confirm resistance, we performed cell viability assays in MCF-7 palbociclib- sensitive cells (MCF-7pS) and MCF-7 palbociclib-resistant cells (MCF-7pR), showing a 10-fold increase in the IC_50_ in the MCF-7pR cells compared to the parental MCF-7pS cells (16.7 vs. 1.8 µM; Figure 1a). To test whether the palbociclib-resistant MCF-7pR cells were cross-resistant to other CDK4/6 inhibitors, we ran cell growth assays with abemaciclib. We confirmed a significant IC_50_ shift in resistant cells compared to parental cells for this latter compound (Figure 1b; IC_50_ 6.8 vs. 0.35 µM). To evaluate the cell growth rate of resistant and sensitive cells, we performed an MTS viability assay over a 96 h time course. The results showed that the proliferation activity of MCF-7pR cells increased almost two-fold at 96 h versus parental cells (Figure 1c).

To investigate the cellular mechanisms leading to resistance against palbociclib, we performed a cell cycle analysis on cells exposed to 1 µM palbociclib for 48 h (Appendix A) and 72 h (Figure 1d,e). In a control medium, sensitive and resistant cells showed comparable G1 populations. On the other hand, as expected [13], we observed a marked G1 arrest in sensitive cells exposed to palbociclib: after 48 h, G1-phase cells comprised 90.7%, while in resistant cells, they comprised only 68.7% (*p* < 0.0001 parental vs. resistant; Appendix A). After 72 h of treatment, the G1 population distribution was 93.9% vs. 70.9% in sensitive and resistant cells, respectively (*p* < 0.0001 parental vs. resistant; Figure 1c). Similarly, the S phase distribution in the absence of drug was slightly lower in the MCF-7pR cells after 48 h (36.3% vs. 21.8%), while there was no difference after 72 h. However, after 48 h of treatment with palbociclib, the MCF-7pS cells showed a marked decrease in the proportion of cells in the S phase compared to the MCF-7pR cells (5.6% vs. 15.5%, *p* value = 0.0062), and after 72 h, the effect was even more evident (MCF-7pS vs. MCF7-pR: 3.4% vs. 14.91%, *p* value = 0.0002). Remarkably, there was virtually no effect of the drug on the S-phase content of the MCF-7pR cells, while the parental cells underwent a > 10-fold reduction (Figure 1e and Appendix A).

To confirm the tumorigenic potential of palbociclib-resistant cells, we analyzed their clonogenicity, showing that the MCF-7pR cells maintained their ability to form new colonies when treated with 1 µM palbociclib, while in the parental cells, colony formation was fully suppressed by the treatment (Figure 1f,g). The total colonies were similar in untreated MCF-7pS and MCF-7pR cells, but when exposed to the drug, colony formation units (CFUs) decreased to zero in sensitive cells, while they were not affected in resistant cells (*p* value = 0.0032, MCF-7pS vs. MCF-7pR after exposure). Moreover, a wound-healing assay showed a higher percentage of wound closure in the resistant line compared to parental cells with or without palbociclib exposure (*p* value < 0.0001, MCF7-pS vs. MC-7pR, treated or not; Appendix A).

Altogether, these data indicate that MCF-7pR cells are not affected by CDK4/6 pharmacological inhibition, and they are a drug-resistant model of HR+/HER2− mLBC.

### 2.2. Pathway Analysis of Palbociclib-Resistant Cells

To explore the molecular mechanisms leading to resistance to palbociclib, we performed a whole-transcriptome analysis of resistant and sensitive cells via RNA-seq.

A total of 1938 genes were found to be differentially expressed in the MCF-7pR vs. MCF-7pS lines (Figure 2a; adjusted Benjamini–Hochberg *p*-value (FDR) < 0.1; |Log_2_-Fold-Change| > 2) and the top differentially genes derived from RNAseq analysis with Log2-Fold-Change > 6 or <−6 are listed in Appendix A. We unexpectedly observed that some genes of the CEACAM family, markers of cancer progression in many solid malignancies [14], were downregulated in MCF-7pR cells (Figure 2a). Our findings seem to corroborate previous results in which CEACAM loss of heterozygosity (LOH) or truncating mutations were associated with tumorigenesis and metastasis in breast cancer [15,16], although this hypothesis needs further study. We also found that pro-apoptotic genes, such as CASP3 and PYCARD, are downregulated in MCF-7pR, whereas the pro-survival BCL2 gene is strongly upregulated (Log2FC = 3.1, padj = 1.15 × 10^−169^; Figure 2a,b), suggesting an imbalance of cell death/survival stimuli in favor of the latter.

A Gene Set Enrichment Analysis (GSEA) performed on the ranked gene list (FDR < 0.25) identified 38 and 12 gene sets enriched in MCF-7pS and MCF-7pR, respectively. Among the differential gene sets (Figure 2b,c) we found a significant enrichment of signatures associated with angiogenesis, the epithelial–mesenchymal transition (EMT), hypoxia and KRAS (enriched in MCF-7pR). The enrichment of the EMT gene set (NES = 1.72, FDR *p* = 0.0) was particularly intriguing as this biological process can lead to resistance and increased invasiveness [17,18]. These data were confirmed using an over-representation analysis in the MCF-7pR cells. In Figure 2d, enriched pathways are shown, with the top terms having the highest significance (FDR ≤ 0.05). This analysis indicated that the highest enrichment ratio was found for TGF-B signaling in thyroid cells for the EMT.

Furthermore, RNA-seq data revealed a marked overexpression of *CDK7*, *CD44*, *VIM*, the ΔN isoform of *p63*, miRNA-210, and *TWIST1*, all of which are implicated in the EMT pathway. A correlation analysis of RNA-Seq and qPCR analyses carried out for 31 DEGs revealed a strong Pearson’s correlation (r = 0.8769), thus providing further validation of the transcriptomic results (Appendix A).

Notably, among the most significantly deregulated genes, CDKN2B emerged with a log_2_ fold-change of −8.04 (adjusted *p* value = 1.07 × 10^−13^). To confirm these data, we evaluated the CDKN2B gene expression via a RT-qPCR and protein expression via Western blotting. In line with the whole-transcriptome data, the RT-qPCR showed 175-fold decreased expression of CDKN2B in the resistant cells (Appendix A; *p* value = 0.0125). This result was confirmed by a 159-fold reduction at the protein level (Figure 2e; *p* value = 0.0021). In addition, we detected a reduction in the p27/Kip1 protein and a marked overexpression of Cyclin E, as already shown by other works [19,20]. We then sought to study the molecular mechanisms potentially responsible for the downregulation of CDKN2B. Interestingly, in the RNA-seq data, we noted a striking increase in the expression of MIR31HG, a precursor of miR-31. Notably, miR-31 has been reported as a negative regulator of the CDKN2B promoter [21,22]. Hence, via a qRT-PCR, we quantified the expression of miR-31-5p and miR-31-3p: the results confirmed the RNA-seq data, showing that miR31-5p was significantly overexpressed in the MCF-7pR cells (*p* value = 0.0073) (Figure 2g). To verify if CDKN2B deletion has a key role in resistance, we silenced the CDKN2B gene in the MCF-7pS cells and evaluated sensitivity to palbociclib in control (scramble shRNA) and CDKN2B-silenced cells and observed no change, indicating that the loss of CDKN2B alone is not sufficient to induce resistance (Appendix A).

### 2.3. Detection of TWIST1 Over-Expression in Palbociclib Resistant MCF-7 Cells

Among top upregulated genes involved in the EMT, *TWIST1* (Figure 3a) attracted our attention as its expression has been linked to high-grade invasive breast carcinoma [23]. An RT-qPCR and protein detection validated the RNA-seq results (Figure 3b,c). Additionally, the MCF-7pR cells displayed classical EMT markers, such as increased CD44 expression, reduced E-Cadherin levels, and elevated N-Cadherin and Vimentin expression levels (Figure 3c,d). These findings indicate that the EMT was occurring in the resistant cells.

To identify a core set of differentially expressed genes likely under the direct transcriptional control of *TWIST1*, we analyzed the intersection between genes bound by *TWIST1* in promoter regions in MCF7 cells (https://www.ncbi.nlm.nih.gov/geo/query/acc.cgi?acc=GSE189826) and DEGs from our RNA-seq dataset (MCF-7pS vs. MCF-7pR). This analysis (false discovery rate (FDR)  <  0.001) revealed a very large number of co-occurring genes (4520/7012 total *TWIST1* promoter peaks; Figure 3e). We selected DEGs by filtering with −1 < Log_2_(fold change) > 1, resulting in 1133 ChIP-identified *TWIST1* target genes. Of these, 695 (61.34%) were upregulated in our dataset. Finally, we generated a stable TWIST1-EGFP MCF-7 cell line (Figure 3f,g) to study the role of TWIST1 in drug resistance. TWIST1-overexpressing MCF-7 cells did not show significant resistance to palbociclib, suggesting that TWIST1 is not the only determinant of resistance. However, the qPCR quantification of a subset of 17 *TWIST1* target genes on TWIST1-overexpressing cells and MCF-7pR cells evidenced a strong correlation (Pearson r = 0.6496, Appendix A) between the two lines, further supporting the evidence of TWIST1 activation in palbociclib-resistant MCF-7pR cells.

### 2.4. Overcoming Resistance in MCF-7pR by Targeting TWIST1

Although TWIST1 alone was not sufficient to induce resistance in parental cells, it may be necessary for resistant cells’ growth. We sought to evaluate the potential of TWIST1 as a therapeutic target to reverse drug resistance. To this end, we took advantage of harmine, a β-carboline alkaloid extracted from plants [24], which degrades TWIST1 protein [25,26]. In particular, TWIST1 degradation is induced in a dose-dependent manner in breast cancer cells and leads to the suppression of migration and invasion. The degradation is proteasome-mediated and does not involve gene regulation or TWIST1 mRNA stability [25]. We examined TWIST1 protein expression after harmine treatment and showed that TWIST1 was reduced by half at 1 µM and by >80% at 5 and 10 µM harmine in MCF-7pR cells compared to an untreated CTRL (Figure 4a,b). Based on these findings, which confirmed the effect of harmine on MCF-7pR cells, a cell viability assay was carried out to assess the activity of harmine in MCF-7pS and pR cells (IC_50_ = 9 vs. 11 µM, Figure 4c) and of its combination with palbociclib. Notably, the treatment of MCF-7pR cells with palbociclib in association with harmine completely reverted palbociclib resistance, showing an IC_50_ equal to that of sensitive cells (Figure 4d,e). We further analyzed drug interaction using the Bliss model: the combination of palbociclib and harmine had a synergistic effect in inhibiting the growth of MCF-7pR cells in three out of four combinations (Figure 4f). On the other hand, in MCF-7pS cells, synergism was only observed with one combination; otherwise, the effect was mostly additive. Appendix A displays normalized proliferation, and the combination of both drugs is compared to the expected additive effect according to Bliss, which is obtained by multiplying the normalized proliferation of single treatments. These results indicate that harmine can blunt resistance to palbociclib in MCF-7 cells, and the combination of the two drugs may be considered to revert or prevent resistant disease. This effect is possibly obtained through TWIST1 protein downmodulation, although this needs further investigation.

## 3. Discussion

Resistance to CDK4/6 inhibitors is a major problem in the treatment of mLBC patients. Although these targeted treatments in combination with endocrine therapy (aromatase inhibitors or fulvestrant) have demonstrated efficacy in prolonging PFS in both endocrine-sensitive and endocrine-resistant patients, disease progression develops in approximately one-quarter of the patients [27,28]. Therefore, it has become an urgent need to elucidate the key regulators of drug resistance and identify biomarkers of resistance. CDKs’ activity is tightly regulated by the INK4 (CDKN2A, 2B, 2C, and 2D) [29] and CIP/KIP protein families (CDKN1A, 1B, 1C) [30]. These CDK-inhibiting proteins have overlapping but not completely redundant activities: in fact, Green et al. reported that a high level of expression of CDKN2A predicts resistance to CDK4/6 inhibitors [31], while Xia et al. showed that CDKN2B is more effective than CDKN2A as a cell-cycle progression inhibitor and its downregulation can lead to tumor initiation [29].

To explore the mechanisms of resistance to CDK4/6 inhibitors, we established a reliable preclinical model of acquired palbociclib resistance. Subsequently, we demonstrated that a palbociclib-resistant cell line is cross-resistant to another CDK4/6 inhibitor, abemaciclib. Cross-resistance among CDK4/6 inhibitors has been reported under both preclinical and clinical conditions [32,33,34]. These data have important clinical implications as patients progressing on palbociclib are predicted to be refractory to alternative CDK 4/6 inhibitors. Our model shows the upregulation of CYCLIN E and RPS6, consistent with findings in CDK4/6I-resistant BC [35,36].

We then explored the gene expression profiles of sensitive and resistant cell lines. In our drug-resistant model, we showed the enrichment of EMT and hypoxia pathway genes and a reduction in G2/M checkpoint genes. In addition, from the analysis, a strong upregulation of TP63 was detected. In this work, we detected in the resistant cells the upregulation of both ΔNp63, which has key role in the EMT [37], and TAp63, an inhibitor of metastasis [38]; however, the level of upregulation differed significantly, so we hypothesize that the ΔNp63/TAp63 ratio, strongly in favor of ΔNp63 in resistant cells, indicates a tumorigenic effect in our model, as Park and colleagues already suggested in cervical cancer cell lines [39]. Remarkably, pro-apoptotic genes that are known to be controlled by TP63 (CASP3 and PYCARD) were downregulated in our model, supporting this hypothesis. In contrast, the anti-apoptotic gene BCL2 was upregulated, pointing to a possible targeted therapy using a selective BCL2 inhibitor such as venetoclax [40].

The observed upregulation of CDK7 may be of interest in this context as CDK7 is a member of the CDK Activating Kinase (CAK) complex, involved in phosphorylation of many CDKs, including CDK4/6, leading to their activation and progression beyond the G1/S checkpoint [41]. Our results suggest CDK7 as a potential therapeutic target to overcome CDK4/6i resistance, in agreement with Guarducci et al. [42]. Further studies are warranted to test CDK7 inhibitors such as THZ1 in this context.

Among the differentially regulated genes, we validated the robust abrogation of *CDKN2B*. CDKN2B, (INK4B, p15INK4B), located in band 9p21, encodes a protein that induces a G1-phase cell cycle arrest by inhibiting CDK4/6 [43].

Of note, differentially expressed microRNA precursors emerged via an RNA-seq analysis in our model. In particular, we found that miR-210-3p was upregulated. According to published data, BC patients have a poorer outcome if elevated levels of miR-210 are present [44]. Moreover, miR-31 is a key regulator of breast cancer invasiveness and metastasis [45,46], as well as a regulator of mammary stem cells [47]. miR-31 has been described to have an oncogenic role in lung, pancreatic, and colon cancers by repressing CDKN2B expression [21].

Another target that emerged by this analysis was *TWIST1* as an important driver of the EMT. We found that our cell model stably expressing TWIST1 showed a proliferation rate and a transcriptomic signature strictly similar to the MCF-7pR cells. In line with these results, we hypothesized that TWIST1 may be a marker of mLBC resistance to CDK4/6 inhibitors and a target to overcome this resistance. Indeed, several studies have investigated a number of molecules as potential TWIST1 inhibitors, including harmine [25,26]. TWIST1 is a post-translational target of harmine that induces protein degradation, as reported in the literature [47]. We showed that in MCF-7pR, TWIST1 protein expression was markedly reduced by harmine and, in addition, the combination of palbociclib and harmine acted synergistically to reverse resistance in MCF-7pR cells, which showed an IC_50_ comparable to MCF-7pS cells when treated in the presence of harmine. Although further analyses are needed to conclusively confirm TWIST1 as a major driver of resistance, this work advocates the use of harmine, or functional analogues, to fight CDK inhibitor resistance.

In conclusion, our results provide new insights into the complex resistance mechanisms associated with the use of CDK4/6 inhibitors. In particular, we report the downregulation of the CDKN2B/p15INK4B CDK-inhibitor protein that may reduce the efficacy of CDK4/6 inhibition. Moreover, the overexpression of miR31, a negative regulator of CDKN2B with a strong oncogenic potential in cancer, is of particular interest. In addition, our results showed that the EMT, in which TWIST1 is one of the master regulators, is one of the most important pathways in cancer cell resistance. This transcription factor could be targeted to provide a new therapeutic option. Further studies are needed to find new drugs to interfere with TWIST1’s activity, e.g., by modifying promoter methylation, mRNA stability, or by developing a new protein degrader.

## 4. Material and Methods

### 4.1. Cell Lines and Selection of Palbociclib Resistant Cells

MCF-7 cells were purchased from the American Type Culture Collection (ATCC, Manassas, VA, USA). The cells were maintained in high-glucose Dulbecco’s modified Eagle medium supplemented with 10% fetal bovine serum (FBS), 1 mM L-glutamine, and 100 units/mL penicillin-streptomycin (Euroclone, Milano, Italy) and grown in a humidified incubator at 37 °C and 5% CO_2_. Palbociclib (PD0332991), isethionate, and abemaciclib (LY2835219) were purchased from SelleckChem (Selleck Chemicals LLC, Houston, TX, USA). A palbociclib stock solution was dissolved in water at a concentration of 1 mM, while abemaciclib was thawed in DMSO at a concentration of 5 mM and stored at −80 °C. Harmine (C_13_H_12_N_2_O) was purchased by Merck (Merk Life Science S.r.l. Milano, Italy), dissolved in DMSO at a concentration of 25 mM, and stored at −20 °C. To select palbociclib-resistant cells, fresh medium with a 500 nM compound was refreshed every 2–3 days, in line with published protocols [48,49]. The concentration was then increased step by step once the cells were able to proliferate and to reach confluence. The resistant cells, herein named MCF-7pR, were established after 12 months of selection and then kept in culture with 1 µM palbociclib. The compound was washed out for 48 h before experiments were performed.

### 4.2. Lentiviral Infection

Two hundred ninety-three 293FT packaging cells (Thermo Fisher Scientific, Waltham, MA, USA) were transfected with 7 µg psPAX2, 1.4 µg pCMV-VSV-G (kind gifts from Prof. Piazza), and 7 µg of TWIST1 (RC202920L4, OriGene Technologies Inc., Rockville, MD, USA), a CTRL vector obtained from p-lenti-puro-mEGFP using jetPRIME (Polyplus-Transfection SA, Illkirch, France). Lentivirus was collected 2 and 3 days after transfection. To generate cells stably infected with lentiviruses, 5 × 10^5^ cells were transduced via spin infection in lentiviral supernatants supplemented with 4 μg/mL polybrene (Merk Life Science S.r.l. Milano, Italy) and 10% high-glucose DMEM. After 48 h, the cells expressing wild-type TWIST1 and scrambled cells were resuspended in complete medium and selected for 3 weeks in the presence of Puromycin, 2 μg/mL.

### 4.3. Growth Inhibition Assay

The cells were seeded into 96-well plates (2500–3000 cells/well) and incubated for 24 h. The growth medium was then replaced with a medium containing palbociclib at different concentrations. The percentage of growth inhibition was quantified after 72 h using a CellTiter-AQueous One Solution Cell Proliferation Assay (Promega, Madison, WI, USA), according to the manufacturer’s instructions. Relative cell viability was obtained by first subtracting the background signal (the CellTiter-AQequos reagent in the medium) and then normalizing vs. untreated control cells. The absorbance was measured using a microplate reader (TECAN, Tecan Trading AG, Männedorf, Switzerland) with an excitation and emission of 490 nm. IC50 was determined using a sigmoidal regression model using GraphPad Prism 6.0 software (GraphPad Software, La Jolla, CA, USA) and was defined as the concentration of a drug required for a 50% reduction in growth. Each experiment was repeated at least three times.

To calculate synergism, the Bliss independence model was applied [50]. Briefly, the expected additive effect of the drug combination was calculated using the Bliss additivity effect as follows: E_AB_ = E_A_ +E_B_ − E_A_E_B_, where E_A_ and E_B_ represent the inhibitory effects of the single drugs A and B. The difference between the expected and the observed effects of the combination is recorded as the Bliss score. If the observed effect is greater than the expected one, the combination is considered synergistic. We report in the results the synergy score, defined as the percent deviation from the Bliss prediction. A synergy score ≥ 10 was considered synergistic.

### 4.4. Cell Cycle Analysis

Parental MCF-7 cells sensitive to palbociclib (MCF-7pS) and MCF-7pR cells were seeded in a 6-well plate. The cells were not synchronized. After 24 h, the cells were treated with 1 µM palbociclib, and at 48 h and 72 h post treatment, the cells were harvested. The experiment was performed in triplicate. For a cell cycle analysis, the cells were harvested and resuspended in a buffer (PBS + 2% FBS), washed with PBS 1×, and fixed with cold 70% ethanol for 30 min on ice. The cells were then washed twice with PBS 1X and stained with propidium iodide 20 µg/mL (P4864, Sigma, St. Louis, MI, USA) and incubated with an RNAse A solution at a final concentration of 10 µg/mL (Merk Life Science S.r.l. Milano, Italy) for 30′ at room temperature, protected from light. The cells were investigated for their DNA content using an Attune™ NxT flow cytometer (Thermo Fisher Scientific, Waltham, MA, USA). The propidium iodide signal was collected using the BL-2 filter (574/26). Samples were prepared in triplicate and analyzed on at least 2 × 104 events per sample. The analysis was performed with De Novo Software FCS Express 7.0.

### 4.5. Clonogenic Survival Assay

Cells (MCF-7pS and MCF-7pR) were seeded at 1500 cells/well in a 6-well plate. After 24 h, the medium was replaced with a control medium (CTRL) or a medium + palbociclib (1 µM) every 72 h. After 14 days, the cells were washed with PBS 1X, fixed with methanol for 10′ at room temperature, and then washed twice with PBS 1X. Then, the cells were stained with 0.5% crystal violet and 25% methanol, and images were captured using a Biorad Instrument camera (Bio-Rad, Hercules, CA, USA) and converted into TIFF images. An analysis was performed using ImageJ software. We calculated the growing area of treated cells normalized versus untreated cells.

### 4.6. Wound Healing Assay

MCF-7pS and MCF-7pR cells were seeded at a concentration of 10^6^ per well in a 6-well plate and grown until they reached 90% confluence. A scratch was made in the wells with a sterile pipette tip, and the cells were cultured in reduced FBS (2%) medium, supplemented or not with palbociclib 1 µM for 48 h. Pictures were taken at 0, 24, and 48 h to measure the migration capacity of the cells. The scratch area was calculated using ImageJ software, and each condition was normalized vs. its own T0 to obtain the percentage of wound closure.

### 4.7. RNA Isolation, RNA-seq and Bioinformatic Analysis

MCF-7pS and MCF-7pR cells were seeded in triplicate at a density of 3 × 10^6^ cells in T75 flasks and incubated to reach 70% confluence. MCF-7pR cells were grown in the absence of palbociclib for 48 h before RNA extraction. Total RNA from the cells was extracted using RNeasy MiniKit (QIAGEN GmbH, Hilden, Germany) according to the manufacturer’s instructions, while we used a miRNAeasy kit (QIAGEN GmbH, Hilden, Germany) to extract total RNA, including microRNAs.

Triplicates of 1 µg RNA samples (MCF-7pS and MCF-7pR) were sent to Galseq srl for polyA selection, library preparation, and paired-end sequencing at approximately 50 million clusters per sample. Sequencing was performed on an Illumina HiSeq instrument in a 2 × 150 bp configuration.

Fastq reads were initially quality checked using FastQC (https://www.bioinformatics.babraham.ac.uk/projects/fastqc/, accessed on 5 August 2023). Subsequently, paired reads were aligned to the human reference genome (GRCh38/hg38) using the splice-aware aligner STAR v2.6.0a22 [51]) and further processed with Samtools [52]. Per-gene read counts were generated, using the STAR quantMode option. Bioconductor DESeq2 package v. 1.30 [53] was used for normalization and a differential expression analysis. Sorted, indexed bam files were used for a manual quality check of the alignment profiles.

### 4.8. Immunoblotting Analysis

Cell lysates from parental and resistant cells were prepared in an RIPA buffer with protease and phosphatase inhibitors, and 100 μg of total proteins was loaded and an SDS-PAGE was conducted with the specific antibodies listed in the table below. The proteins were blotted on a nitrocellulose membrane and blocked in 5% non-fat milk. The incubation of the primary antibody occurred for 2 h at room temperature or overnight at 4 °C. The chemiluminescence signal was detected using a ChemiDoc XRS+ Imaging System (Bio-Rad, Hercules, CA, USA).

### 4.9. Quantitative Reverse-Transcription Polymerase Chain Reaction (RT-qPCR)

Total RNA (2 µg) was retrotranscribed with M-MLV Supermix 5X (GeneSpin, Milan, Italy) in a 20 µL reaction volume. Of the resulting first-strand cDNA, 2 µL were amplified with QMastermix 2X (GeneSpin, Milan, Italy) in triplicate using a StepOnePlus™ Real-Time PCR System. Relative expression was normalized to the GAPDH using the 2^−ΔΔCt^ method [54]. Furthermore, to analyse microRNA expression levels, we used TaqMan^®^ Advanced miRNA Assays (Thermo Fisher Scientific), following the protocol instructions. Ten nanograms of total RNA were retrotranscribed per reaction. Three independent biological replicates were performed. The TAQMAN assays used are listed below (Table 1 and Table 2).

### 4.10. Statistical Analysis

Statistical analyses were performed using GraphPad Prism 6 software. Quantitative data, collected from independent experiments, were expressed as the mean ± standard error of the mean (SEM). Differences between two data sets were determined by a two-sided *t*-test or an unpaired *t* test with Welch’s correction or the Mann–Whitney test. Furthermore, the wound-healing results were analyzed using a two-way Anova and Bonferroni tests for multiple comparisons. *p*-values were reported and considered significant when below the nominal 0.05 significance level.

## Figures and Tables

**Figure 1 ijms-24-16294-f001:**
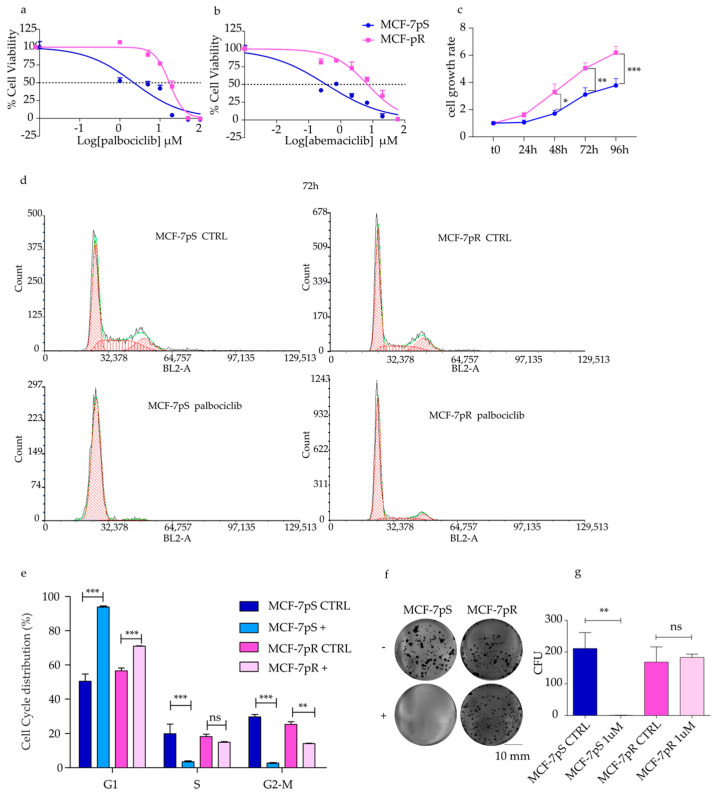
Characterization of a palbociclib-resistant MCF7 cell line. (**a**,**b**) Dose-response curves in MCF-7pS and MCF-7pR cells showing the effect of treatment with varying concentrations of palbociclib and abemaciclib. Dashed lines denote IC_50_ values. (**c**) Cell growth rate of MCF-7pR vs. MCF-7pS. (**d**,**e**) Representative images of DNA content of a cell cycle analysis performed by flow cytometry to examine the effect of palbociclib treatment (1 μM) on MCF-7pS and MCF-7pR cells at 72 h (red line indicates diploid G1, S and G2; green is fit line) and a representative DNA histogram of one experiment. (**f**,**g**) Clonogenic assay showing the effect of 1 μM palbociclib treatment on the proliferation of MCF-7pS and MCF-7pR cells. Representative images of clonogenic assays CTRL and palbociclib treatment (+) (**f**) and histograms reporting colony formation units (CFUs) (**g**). Scale bar is equal to 10 mm. Error bars describe the SEM, and an unpaired *t* test determined *p*-values: * *p* < 0.05, ** *p* < 0.01, *** *p* < 0.001, and ns—not significant.

**Figure 2 ijms-24-16294-f002:**
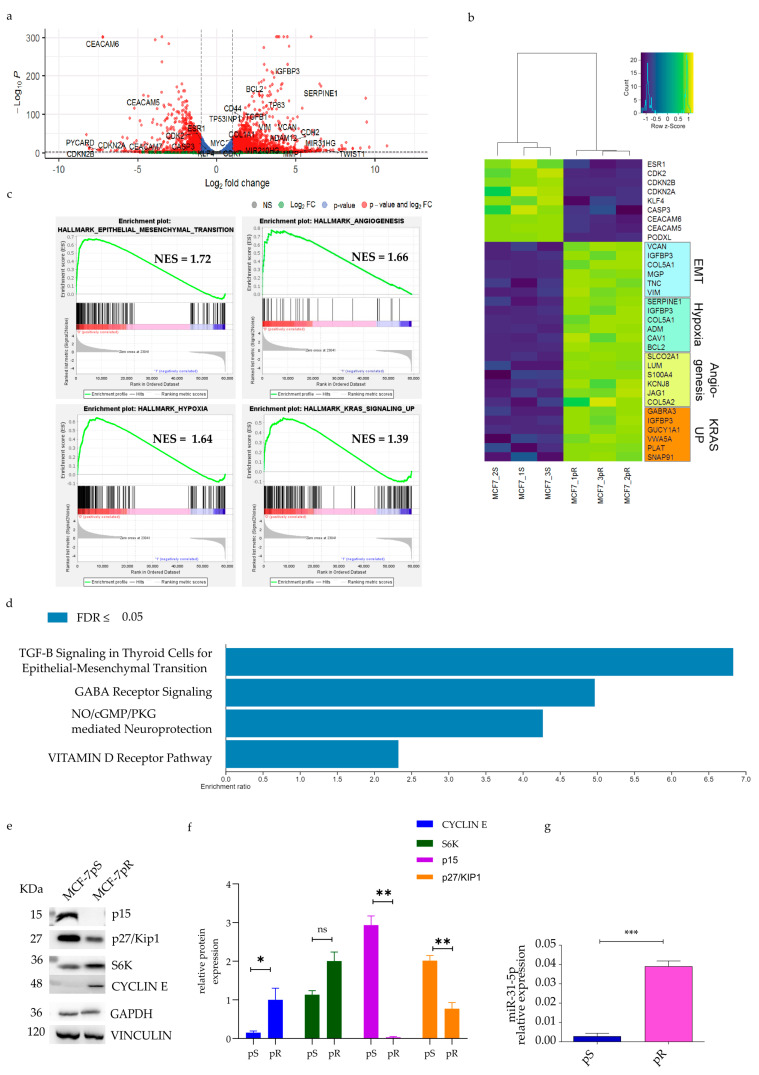
Transcriptomic analysis of palbociclib-resistant vs. palbociclib-sensitive MCF7 cells. An RNA-seq analysis was performed on MCF-7pR vs. MCF-pS cells. Note that the MCF-7pR cells represent cells that were maintained in 1 μM palbociclib without interruption, and a wash-out of 48 h was performed before the RNA extraction. (**a**) Volcano plots. Each point in the volcano plot represents one gene. The blue and red dots represent statistically significant differentially expressed genes (DEGs). Red dots highlight those genes also by characterized by |Log2-Fold-Change| > 1. Gray and green dots represent unchanged genes characterized by |Log2-Fold-Change| < and >1, respectively. The horizontal and vertical dotted red lines indicate an adjusted Benjamini–Hochberg *p*-value (FDR) < 0.1; |Log2-Fold-Change| > 1. (**b**) Hierarchical cluster heatmaps; (**c**) GSEA enrichment plots of the epithelial–mesenchymal transition (EMT), hypoxia, angiogenesis, and KRAS up. (**d**) Bar plot showing the enriched pathways for the genes upregulated in palbociclib-resistant cells. The *x*-axis indicates the fold enrichment for each pathway term. Ordering is based on significance, with the top terms having the highest significance (FDR ≤ 0.05). (**e**) Representative images of Western blotting and (**f**) quantification of target proteins from four independent experiments. (**g**) A qPCR analysis shows an increase in miR-31-5p as a possible inhibitor of CDKN2B. For all graphs, error bars describe SEMs, and an unpaired *t* test determined *p*-values: ns not significant, * *p* < 0.05, ** *p* < 0.01, and *** *p* < 0.001.

**Figure 3 ijms-24-16294-f003:**
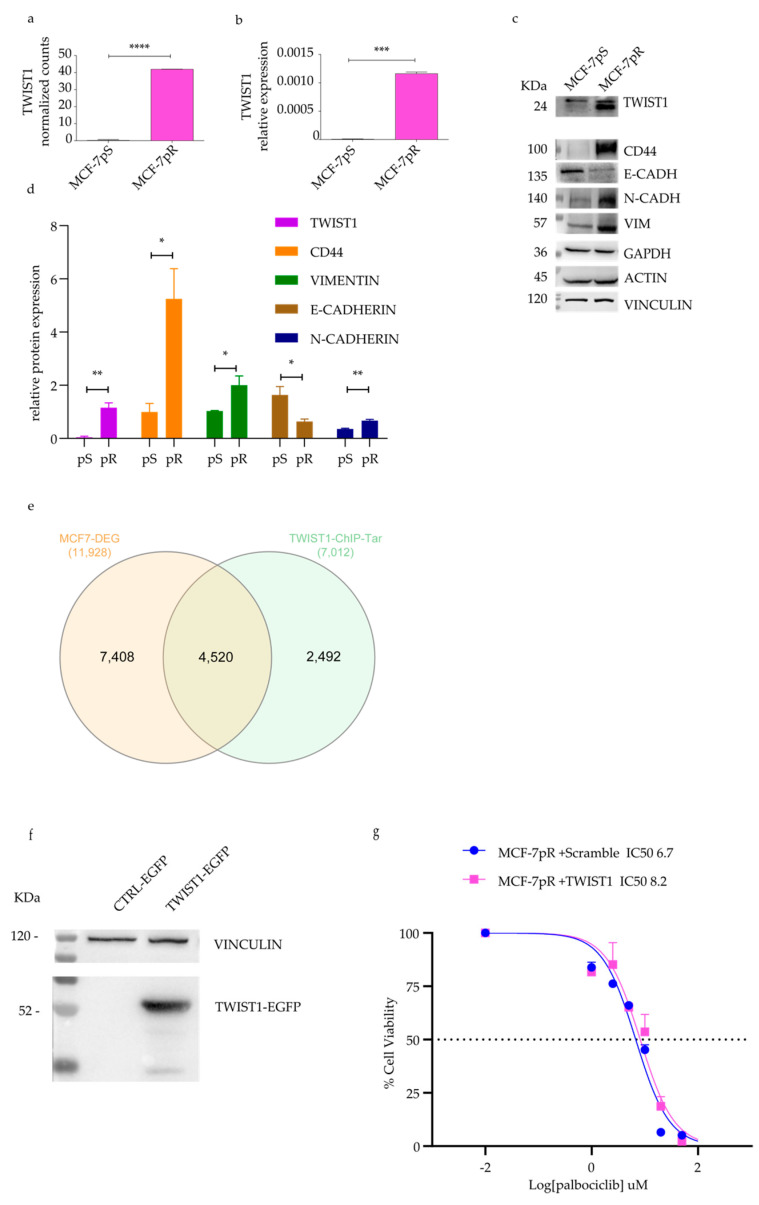
The overexpression of TWIST1 in MCF7-pR cells and an evaluation of other EMT markers. (**a**) RNA-seq results showing TWIST1 normalized counts. (**b**) A qPCR analysis shows the downregulation of the TWIST1 mRNA level. (**c**) Representative image of Western blotting and (**d**) the quantification of three independent experiments. For all graphs, error bars describe SEM, and an unpaired *t* test determined *p*-values: * *p* < 0.05, ** *p* < 0.01, *** *p* < 0.001, and **** *p* < 0.0001 (**e**) Venn diagram showing the number of differentially expressed genes (orange circle) being directly bound by TWIST1 within their promoter region (green circle). (**f**) The MCF-7 cell line was analyzed using WB, showing the stable expression of TWIST1-EGFP, while (**g**) dose–response curves were generated to show the effect of varying concentrations of palbociclib on CTRL-EGFP and TWIST1-EGFP MCF-7pS cells. The IC50 values are denoted by dashed lines.

**Figure 4 ijms-24-16294-f004:**
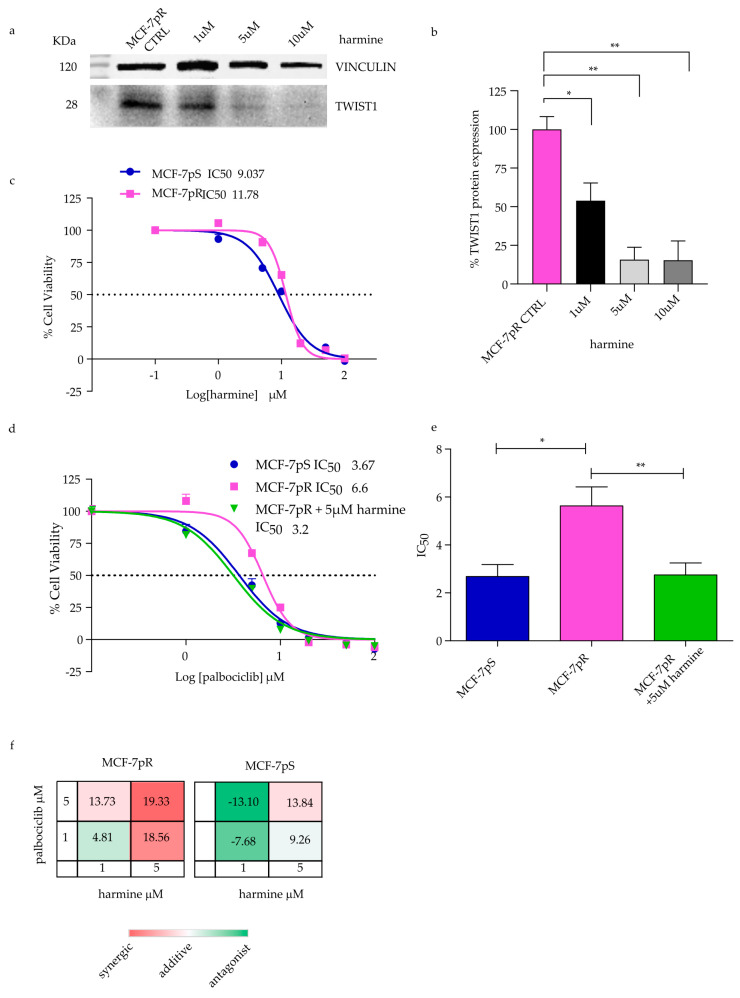
Inducing TWIST1 degradation with dose-dependent harmine and overcoming palbociclib resistance. (**a**) Representative image of Western blotting and (**b**) the quantification of three independent experiments. (**c**) Dose–response curves of MCF-7pS and pR cells treated with harmine. (**d**) Dose–response curves for MCF-7pS and MCF-7pR cells showing the effect of treatment with varying concentrations of palbociclib and harmine at 5 µM. Dashed lines depict IC50 values. The histogram reports the mean IC50 values ± the standard deviations (SEMs) for the cell lines MCF-7pS and pR treated with palbociclib and for MCF-7pR with palbociclib in association with harmine 5 µM for 72 h in triplicate (**e**). Unpaired *t*-tests determined the *p*-values: * *p* < 0.05 and ** *p* < 0.01. (**f**) Bliss synergy plot of combined palbociclib and harmine. A score between −10 and +10 is considered additive. Red: synergy; white: additivity; green: antagonist effect.

**Table 1 ijms-24-16294-t001:** Antibodies used in the WB analysis.

Antobody	Type	Clone	Code	Origin	Brand	Dilution	Milk/BSA
*N-CADHERIN*	Primary	H-2	sc-393933	Mouse	Santa Cruz	1:200	BSA 5%
*E-CADHERIN*	Primary	G-10	sc-8426	Mouse	Santa Cruz	1:400	BSA 5%
*HCAM (CD44)*	Primary	EPR18668	ab189524	Rabbit	Abcam	1:200	BSA 5%
*RPS6*	Primary	C-8	sc-74459	Mouse	Santa Cruz	1:400	MILK 5%
*TWIST1*	Primary	2C1a	sc-81417	Mouse	Santa Cruz	1:800	BSA 5%
*p15 (CDKN2B)*	Primary	D-12	sc-271791	Mouse	Santa Cruz	1:500	MILK 5%
*CYCLIN-E*	Primary	HE12	sc-247	Mouse	Santa Cruz	1:200	MILK 5%
*VIMENTIN*	Primary	D21H3	5741S	Rabbit	Cell Signaling	1:1000	BSA 5%
*p27/KIP1*	Primary	F-8	sc-1641	Mouse	Santa Cruz	1:400	MILK 5%
*β-ACTIN*	Primary	AC-15	A1978	Mouse	SIGMA	1:2000	BSA 5%
*VINCULIN*	Primary	V284	05-386	Mouse	Millipore	1:4000	BSA 5%/MILK 5%
*GAPDH*	Primary	0411	sc-47724	Mouse	Santa Cruz	1:3000	BSA 5%/MILK 5%
*anti-MOUSE*	Secondary	IgG (H/L)	170-6516	Goat	BIORAD	1:2000	BSA 5%
*anti-RABBIT*	Secondary	IgG (H/L)	170-6515	Goat	BIORAD	1:2000	BSA 5%

**Table 2 ijms-24-16294-t002:** Assays ID list. Displayed here are the assays ID used in the quantitative PCR experiments. The list includes the corresponding gene symbols for each assay ID.

Assay ID	Gene Symbol
Hs01046816_m1	Esr1
Hs00958111_m1	Vim
Hs01556702_m1	Pgr
Hs00944025_m1	Ceacam5
Hs00161904_m1	Snai2
Hs01047973_m1	Runx2
Hs00998133_m1	Tgfb1
Hs00609133_m1	Col5a1
Hs00153133_m1	Ptgs2
Hs00181211_m1	Igfbp3
Hs01053790_m1	Abcg2
Hs00602051_mH	Fscn1
Hs00187067_m1	Nr5a2
Hs00899658_m1	Mmp1
Hs03988977_m1	Ceacam7
Hs00167155_m1	Serpine1
Hs00171642_m1	Vcan
Hs00178811_m1	Ntrk2
Hs01115665_m1	Tnc
Hs01041212_m1	Bhlhe40
Hs99999018_m1	Bcl2
Hs00924091_m1	Cdkn2a
Hs00179899_m1	Mgp
Hs00203118_m1	Pycard
Hs99999905_m1	Gapdh
Hs00793225_m1	Cdkn2b
Hs01075861_m1	CD44
Hs00361486_m1	Cdk7
Hs00358836_m1	Klf4
Hs00978339_m1	ΔNp63
478012_mir	hsa-miR-31-3p
478015_mir	hsa-miR-31-5p
477970_mir	hsa-miR-210-3p
477860_mir	hsa-miR-16-5p

## Data Availability

Data are available upon request to nicoletta.cordani@unimib.it. The RNA-seq data were deposited in NCBI’s Sequence Read Archive and are available at the following link: https://www.ncbi.nlm.nih.gov/sra/PRJNA798002, November 2023.

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
