# Peer review of "TWIST1 Upregulation Is a Potential Target for Reversing Resistance to the CDK4/6 Inhibitor in Metastatic Luminal Breast Cancer Cells"

_ijms, 2023, doi:10.3390/ijms242216294_

Round 1

Reviewer 1 Report

Comments and Suggestions for Authors

In this manuscript, the authors performed a series of experiments to study the drug resistance in metastatic luminal breast cancer cells. The idea is pretty interesting. However, the provided results are still preliminary, and some concerns need to be addressed well.

1.     The authors missed a great chance to improve the novelty and integrity in this study. From RNAseq, the authors identified TWIST1 is a potential target of drug resistance for CDK4/6 inhibitors. TWIST1 is well known as a key regulator for EMT regulation. The authors should provide some key experiments to show if EMT process has been changed between sensitive cells and resistant cells. Instead, the authors performed a lot of less meaningful analysis, such as Fig 4/5 and Fig 8b/c.

2.     In Fig.7, the authors overexpressed TWIST1 in sensitive cells. They should check if TWIST1 overexpression can change the drug resistance, instead of only looking at proliferation.

3.     In part 2.4, the authors did not get any conclusions and also no discussion for the results. I cannot tell any differences in Fig.7b, but the authors showed statistical significance in Fig 7c. Please provide the details for quantification.

4.     Fig 1c missed the labeling of cell cycle phases on X axis.

5.     To evaluate proliferation effect, the authors should directly check cell growth rate except Ki67 staining.

Comments on the Quality of English Language

The English is pretty good overall ,and only have minor errors.

Author Response

  • Comments and Suggestions for Authors

In this manuscript, the authors performed a series of experiments to study the drug resistance in metastatic luminal breast cancer cells. The idea is pretty interesting. However, the provided results are still preliminary, and some concerns need to be addressed well.

The authors missed a great chance to improve the novelty and integrity in this study. From RNAseq, the authors identified TWIST1 is a potential target of drug resistance for CDK4/6 inhibitors. TWIST1 is well known as a key regulator for EMT regulation. The authors should provide some key experiments to show if EMT process has been changed between sensitive cells and resistant cells. Instead, the authors performed a lot of less meaningful analysis, such as Fig 4/5  and Fig 8b/c.

Author response: Thank you, for your input. In response, we conducted some WB analysis to demonstrate the changes in the EMT process. These results are presented in Figure 3. Following the reviewer’s suggestion, the whole manuscript has been rearranged: figures 4 and 8b-c have been moved to supplementary, while figure 5 was excluded.

In Fig.7, the authors overexpressed TWIST1 in sensitive cells. They should check if TWIST1  overexpression can change the drug resistance, instead of only looking at proliferation .

Author response: Thank you, for your advice. You were right, it was better to insert the results in main figures (now Figure 3f and g).

In part 2.4, the authors did not get any conclusions and also no discussion for the results. I cannot tell any differences in Fig.7b, but the authors showed statistical significance in Fig 7c. Please provide the details for quantification.

Author response: Thank you for your input. We removed Ki-67 data, as they did not provide conclusive results.

Fig 1c missed the labeling of cell cycle phases on X axis.

Author response: Thank you, for your comment. We inserted the labels of cell cycle phases on X axis.

To evaluate proliferation effect, the authors should directly check cell growth rate except   Ki67 staining, fig ki67 in supplementary

Author response: Thank you, for your input. We check the cell growth rate with MTS assay at T0, T24h, T48h, T72h, and T96h for sensitive and resistant cells and it confirmed Ki-67 staining. We decided to show only the cell growth rate (Figure 1c).

Reviewer 2 Report

Comments and Suggestions for Authors

This is an overall well-conducted study of induced resistance to palbociclib in a breast cancer cell line. The findings in studies such as these are important for thinking about how to address acquired resistance to targeted treatment.

The strengths in this study include

1. There is a comprehensive and well-thought out series of experiments to address several aspects of cancer growth and biology. 

2. Experiments were well-described and data was presented that overall support the main thesis of the manuscript.

3. A new potential target was identified in resistance breast cancer cells.

However, some questions arise with respect to the potential for improvement in the manuscript. These include

1. In the introduction and discussion, other studies designed to evaluate resistance have been conducted. However, little detail is given in either section. Specifically, little detail is given with respect to cyclin E in many other studies in addition to MTOR and PI3K pathways. While they may not have stood out here, a more extensive discussion should be presented to understand why those findings were not seen in this study.

2. MCF-7 cells are standard. Please explain the merits of this cell line versus others. Other studies have induced palbociclib resistance in this cell line. Describe those and why this study is much different.

3. Why was the wound healing assay included? Please explain how this experiment ties into the others. 

4. It appears that TWIST has one have the highest RNA expression values but not the highest. It is unclear what the exact highest and lowest expressors are versus an array of potentially interesting high and low expressors. A list, or confirmation of a list, of the highest and lowest versus those thought to be most biologically interesting and applicable to treatment would be helpful.

5. In some areas of values are given with decimals, sometimes a comma or sometimes a period is used. This should be standardized.

6. Figures 1C and 3C are hard to see and identfy details. 

7. With respect to combination experiments with TWIST, synergy was allude to. However, a model, such as the Bliss or the Chou-Talaly model, was not conducted. To demonstrate synergy, additive, or antagonistic effects, one of these should be conducted with varying concentrations of each drug. 

Comments on the Quality of English Language

Once there is some editing of mostly punctuation and some grammar, it will be adequately readable in final form.

Author Response

Comments on the Quality of English Language

The English is pretty good overall ,and only have minor error

  • Comments and Suggestions for Authors

This is an overall well-conducted study of induced resistance to palbociclib in a breast cancer cell line. The findings in studies such as these are important for thinking about how to address acquired resistance to targeted treatment.

The strengths in this study include

  1. There is a comprehensive and well-thought out series of experiments to address several aspects of cancer growth and biology. 
  2. Experiments were well-described and data was presented that overall support the main thesis of the manuscript.
  3. A new potential target was identified in resistance breast cancer cells.

However, some questions arise with respect to the potential for improvement in the manuscript. These include

In the introduction and discussion, other studies designed to evaluate resistance have been conducted. However, little detail is given in either section. Specifically, little detail is given with respect to cyclin E in many other studies in addition to MTOR and PI3K pathways. While they may not have stood out here, a more extensive discussion should be presented to understand why those findings were not seen in this study.

Author response: Thank you for your feedback. We conducted a Western Blot analysis to confirm the overexpression of Cyclin E and activation of mTOR pathway in resistant cells (figure 2e-f). We have also added additional information on the mTOR and PI3K pathways in the discussion section. While these pathways were not the primary focus of our work, we did investigate their roles in the EMT process.

MCF-7 cells are standard. Please explain the merits of this cell line versus others. Other studies have induced palbociclib resistance in this cell line. Describe those and why this study is much different.

Author response:  Thank you for your feedback. We induced resistance to palbociclib in two cell lines, MCF-7 and T47D, both of which are positive for progesterone receptor and hormone receptor, and negative for HER2. We observed a stronger resistance in MCF-7, which we then chose as our model. Despite previous extensive research on CDK inhibitors resistance, we decided to investigate this resistant model to identify new drug targets, and novel treatments, for the resistant disease.

Why was the wound healing assay included? Please explain how this experiment ties into the others. 

Thank you for your feedback. The wound healing capacity in resistant cells is a characteristic of aggressiveness. This is because the epithelial to mesenchymal transition (EMT) plays a major role in wound healing and can explain some of its pathological aspects.

 It appears that TWIST has one have the highest RNA expression values but not the highest. It is unclear what the exact highest and lowest expressors are versus an array of potentially interesting high and low expressors. A list, or confirmation of a list, of the highest and lowest versus those thought to be most biologically interesting and applicable to treatment would be helpful.

Thank you for your comment. As requested, we added a supplementary table with the top DEGs. Differential genes were inspected and used for pathway analyses. We then focused on TWIST1 based on GSEA outcomes, which revealed that EMT was one of the most significantly enriched gene sets. As Twist1 is a critical transcription factor in the EMT process, we decided to prioritize its investigation.

In some areas of values are given with decimals, sometimes a comma or sometimes a period is used. This should be standardized.

Author response: Thank you for your feedback. We corrected them.

Figures 1C and 3C are hard to see and identfy details. 1c

Author response: Thank you for your input. We corrected them.

With respect to combination experiments with TWIST, synergy was allude to. However, a model, such as the Bliss or the Chou-Talaly model, was not conducted. To demonstrate synergy, additive, or antagonistic effects, one of these should be conducted with varying concentrations of each drug. 

Author response: Thank you, for your input. We calculated synergy using the Bliss model. See Figure 4 and Figure S5.

Reviewer 3 Report

Comments and Suggestions for Authors

The authors in the present article have highlighted in vitro study of the characterization of genes y RNA sequencing associated with CDK4/6 resistance by generating the resistant HR+ cell line MCF-7 cell line using chronic exposure to increasing concentrations of palbociclib compared to sensitive partner. As Twist1 was found to be over-expressed in palbociclib-resistant MCF-7 cells, the authors suggest it as an interesting therapeutic target to overcome CDK4/6i resistance.

The study was well established, experimentally sound.

Some points that need to be worked on:

1.      The figures are too much disorganized with a total of 9 figures. The authors must try to assemble/concise the figures in an organized manner with subheadings and reduce the total figure content. It takes a lot of time to go after pages to find the figure to read with the result section.

2.      Figure 1 c miss the labelling for the cell cycle stages in the graph.

3.      Are the cells synchronized priorly before treatment? The materials and method section didn’t mention on that. Why the authors have chosen 72hrs for cell cycle analysis and didn’t show the 24hrs time point.  

4.      In figure 5 d, the authors should also look for staining against Ki67, TWIST1, and other EMT markers to show as a measure of co-relation with CDKN2B in patient tumor on basal disease and after CDK4/6i therapy.

5.      Show the quantification plots for all the IHC staining done with significance.

6.      In figure 7 b, the immunofluorescence data on TWIST-EGFP and Ki67 staining doesn’t match with the conclusion suggested. Rather the Ki67 signal showed reduced expression in the overexpressed cells. The authors should provide higher resolution images and should show co-localization of GFP+/Ki67+ in the same cells for validation.

7.      Stating on the synergy between the combination studies on harmine treatment and palbociclib in the resistant cell line vs sensitive is a bit overstated. The authors should include more groups of treatment like harmine mono-treatment in resistant vs sensitive, palbociclib mono-treatment in resistant vs sensitive and the combination treatment also in both resistant vs sensitive.

8.      Calculate the IC50 for all the treatments and show the combination index values to understand synergistic effect in the cells.

9.      Since the whole study focusses on EMT induction but there is no data for protein level changes of the known EMT markers in the cells. The authors should try to validate their study including the protein level changes of the EMT markers compared to the only RNA level changes shown by sequencing data analysis.

10.  The authors should try to knock down TWIST1 in the cells and look for sensitization of the cells to palbociclib treatment in the resistant model (in addition to harmine treatment) to validate the results.

Comments on the Quality of English Language

Minor editing is required

Author Response

The authors in the present article have highlighted in vitro study of the characterization of genes y RNA sequencing associated with CDK4/6 resistance by generating the resistant HR+ cell line MCF-7 cell line using chronic exposure to increasing concentrations of palbociclib compared to sensitive partner. As Twist1 was found to be over-expressed in palbociclib-resistant MCF-7 cells, the authors suggest it as an interesting therapeutic target to overcome CDK4/6i resistance.

The study was well established, experimentally sound.

Some points that need to be worked on:

 The figures are too much disorganized with a total of 9 figures. The authors must try to assemble/concise the figures in an organized manner with subheadings and reduce the total figure content. It takes a lot of time to go after pages to find the figure to read with the result section.

Author response: Thank you, for your feedback. The results were reorganized and are now presented in four figures and 5 Supp. Figures.

Figure 1 c miss the labelling for the cell cycle stages in the graph.

 Author response: Thank you, for your comment. We inserted the labels of cell cycle phases on X axis.

Are the cells synchronized priorly before treatment? The materials and method section didn’t mention on that. Why the authors have chosen 72hrs for cell cycle analysis and didn’t show the 24hrs time point.  

Author response: Thank you, for your input. The cells were not synchronized. This information has now been added to Methods. We chose 48h (Figure S1) and 72h (Figure 1d-e) because at 24h palbociclib doesn’t show any effect in MTS assay. We show 72h in figure 1 to correlate with the IC50 results.

In figure 5 d, the authors should also look for staining against Ki67, TWIST1, and other EMT markers to show as a measure of co-relation with CDKN2B in patient tumor on basal disease and after CDK4/6i therapy. Show the quantification plots for all the IHC staining done with significance.

      Author response: Thank you, for your input. Unfortunately, we did not have any biopsy material left to perform IHC. The pathology score for CDKN2B IHC was ++ at basal and + at PD. However, because we decided to reduce the number of figures, we eliminated also IHC because it was only one patient.

In figure 7 b, the immunofluorescence data on TWIST-EGFP and Ki67 staining doesn’t match with the conclusion suggested. Rather the Ki67 signal showed reduced expression in the overexpressed cells. The authors should provide higher resolution images and should show co-localization of GFP+/Ki67+ in the same cells for validation.

Thank you for your contribution. As two reviewers contested KI-67 results, we decided to remove these data altogether, as they did not provide conclusive results.

 Stating on the synergy between the combination studies on harmine treatment and palbociclib in the resistant cell line vs sensitive is a bit overstated. The authors should include more groups of treatment like harmine mono-treatment in resistant vs sensitive, palbociclib mono-treatment in resistant vs sensitive and the combination treatment also in both resistant vs sensitive. Calculate the IC50 for all the treatments and show the combination index values to understand synergistic effect in the cells.

Author response: Thank you, for your input. We included single and combined treatments in sensitive and resistant cells and analysed synergy in a 2x2 concentration matrix (1uM, 5uM). These are the most relevant doses of harmine and palbociclib in these cells. The data were analysed by Bliss method and the results are shown in Figure 4 and Suppl. Fig. 5.

Since the whole study focusses on EMT induction but there is no data for protein level changes of the known EMT markers in the cells. The authors should try to validate their study including the protein level changes of the EMT markers compared to the only RNA level changes shown by sequencing data analysis.

      Author response: Thank you, for your input. In response, we conducted some WB analysis to demonstrate the changes in the EMT process (Figure 3).

The authors should try to knock down TWIST1 in the cells and look for sensitization of the cells to palbociclib treatment in the resistant model (in addition to harmine treatment) to validate the results.

      Author response: Thank you for your insightful suggestion. We attempted to obtain a stably TWIST1-silenced MCF-7pR cell line without success. After consultation with the Editor, we decided to resubmit the manuscript without these data. Consequently, we rephrased our conclusions on the role of TWIST1 in drug resistance in a milder statement.

Round 2

Reviewer 1 Report

Comments and Suggestions for Authors

Thanks. All my concerns's have been well addressed.

Comments on the Quality of English Language

English is good.

Author Response

Thank you very much for your review.